# Multi-omic rejuvenation of human cells by maturation phase transient reprogramming

**Diljeet Gill[1], Aled Parry[1], Fátima Santos[1], Hanneke Okkenhaug[2], Christopher D Todd[1], Irene Hernando-Herraez[1], Thomas M Stubbs[3], Inês Milagre[4]*, Wolf Reik[1,5,6]*†**

[1]Epigenetics Programme, Babraham Institute, Cambridge, United Kingdom; [2]Imaging Facility, Babraham Institute, Cambridge, United Kingdom; [3]Chronomics Limited, London, United Kingdom; [4]Laboratory for Epigenetic Mechanisms/Chromosome Dynamics Lab, Instituto Gulbenkian de Ciência, Oeiras, Portugal; [5]Wellcome Trust Sanger Institute, Hinxton, Cambridge, United Kingdom; [6]Centre for Trophoblast Research, University of Cambridge, Cambridge, United Kingdom

**Abstract** Ageing is the gradual decline in organismal fitness that occurs over time leading to tissue dysfunction and disease. At the cellular level, ageing is associated with reduced function, altered gene expression and a perturbed epigenome. Recent work has demonstrated that the epigenome is already rejuvenated by the maturation phase of somatic cell reprogramming, which suggests full reprogramming is not required to reverse ageing of somatic cells. Here we have developed the first "maturation phase transient reprogramming" (MPTR) method, where reprogramming factors are selectively expressed until this rejuvenation point then withdrawn. Applying MPTR to dermal fibroblasts from middle-aged donors, we found that cells temporarily lose and then reacquire their fibroblast identity, possibly as a result of epigenetic memory at enhancers and/or persistent expression of some fibroblast genes. Excitingly, our method substantially rejuvenated multiple cellular attributes including the transcriptome, which was rejuvenated by around 30 years as measured by a novel transcriptome clock. The epigenome was rejuvenated to a similar extent, including H3K9me3 levels and the DNA methylation ageing clock. The magnitude of rejuvenation instigated by MPTR appears substantially greater than that achieved in previous transient reprogramming protocols. In addition, MPTR fibroblasts produced youthful levels of collagen proteins, and showed partial functional rejuvenation of their migration speed. Finally, our work suggests that optimal time windows exist for rejuvenating the transcriptome and the epigenome. Overall, we demonstrate that it is possible to separate rejuvenation from complete pluripotency reprogramming, which should facilitate the discovery of novel anti-ageing genes and therapies.

*For correspondence:
imilagre@igc.gulbenkian.pt (IM);
wolf.reik@babraham.ac.uk (WR)

Present address: †Altos Labs
Cambridge Institute, Cambridge,
United Kingdom

Competing interest: See page
19

Reviewing Editor: Jessica K
Tyler, Weill Cornell Medicine,
United States

## Editor's evaluation

This study describes a novel "maturation phase transient reprogramming" (MPTR) method to restore the epigenome of cells to a more youthful state. The authors demonstrate the effectiveness of the method to reverse several age-related changes including remodeling of the transcriptome. The method performs favorably compared to other transient reprogramming protocols, and the study will be of interest to developmental biologists as well as researchers who study ageing.

## Introduction

Aging is the gradual decline in cell and tissue function over time that occurs in almost all organisms, and is associated with a variety of molecular hallmarks such as telomere attrition, genetic instability, epigenetic and transcriptional alterations, and an accumulation of misfolded proteins (*López-Otín et al., 2013*). This leads to perturbed nutrient sensing, mitochondrial dysfunction, and increased incidence of cellular senescence, which impacts overall cell function and intercellular communication, promotes exhaustion of stem cell pools, and causes tissue dysfunction (*López-Otín et al., 2013*). The progression of some aging related changes, such as transcriptomic and epigenetic ones, can be measured highly accurately and as such they can be used to construct "aging clocks" that predict chronological age with high precision in humans (*Hannum et al., 2013*; *Horvath, 2013*; *Peters et al., 2015*; *Fleischer et al., 2018*) and in other mammals (*Stubbs et al., 2017*; *Thompson et al., 2017*; *Thompson et al., 2018*). Since transcriptomic and epigenetic changes are reversible at least in principle, this raises the intriguing question of whether molecular attributes of aging can be reversed and cells phenotypically rejuvenated (*Rando and Chang, 2012*; *Manukyan and Singh, 2012*).

Induced pluripotent stem cell (iPSC) reprogramming is the process by which almost any somatic cell can be converted into an embryonic stem cell-like state. Intriguingly, iPSC reprogramming reverses many age-associated changes, including telomere attrition and oxidative stress (*Lapasset et al., 2011*). Notably, the epigenetic clock is reset back to approximately 0, suggesting reprogramming can reverse aging associated epigenetic alterations (*Horvath, 2013*). However, iPSC reprogramming also results in the loss of original cell identity and therefore function. By contrast, transient reprogramming approaches where the Yamanaka factors (Oct4, Sox2, Klf4, and c-Myc) are expressed for short periods of time may be able to achieve rejuvenation without loss of cell identity. Reprogramming can be performed in vivo (*Abad et al., 2013*), and indeed, cyclical expression of the Yamanaka factors in vivo can extend lifespan in progeroid mice and improves cellular function in wild-type mice (*Ocampo et al., 2016*). An alternative approach for reprogramming in vivo also demonstrated reversal of aging-associated changes in retinal ganglion cells and was capable of restoring vision in a glaucoma mouse model (*Lu et al., 2020*). More recently, in vitro transient reprogramming has been shown to reverse multiple aspects of aging in human fibroblasts and chondrocytes (*Sarkar et al., 2020*). Nevertheless, the extent of epigenetic rejuvenation achieved by previous transient reprogramming methods has been modest (~3 years) compared to the drastic reduction achieved by complete iPSC reprogramming. A more detailed comparison of previous methods is provided in *Supplementary file 1*. Here, we establish a novel transient reprogramming strategy where Yamanaka factors are expressed until the maturation phase (MP) of reprogramming before abolishing their induction (maturation phase transient reprogramming, MPTR), with which we were able to achieve robust and very substantial rejuvenation (~30 years) whilst retaining original cell identity overall.

## Results

### Transiently reprogrammed cells reacquire their initial cell identity

Reprogramming can be divided into three phases: the initiation phase (IP) where somatic expression is repressed and a mesenchymal-to-epithelial transition occurs; the MP, where a subset of pluripotency genes becomes expressed; and the stabilization phase (SP), where the complete pluripotency program is activated (*Samavarchi-Tehrani et al., 2010*; *Figure 1A*). Previous attempts at transient reprogramming have only reprogrammed within the IP (*Ocampo et al., 2016*; *Sarkar et al., 2020*). However, reprogramming further, up to the MP, may achieve more substantial rejuvenation. To investigate the potential of MPTR to reverse aging phenotypes, we generated a doxycycline-inducible polycistronic reprogramming cassette that encoded *Oct4, Sox2, Klf4, c-Myc,* and GFP. By using a polycistronic cassette, we could ensure that individual cells were able to express all four Yamanaka factors. This reprogramming cassette was capable of generating iPSC lines from human fibroblasts and induced a substantial reduction of DNA methylation age throughout the reprogramming process, consistent with previous work using a different reprogramming system (*Olova et al., 2019*; *Figure 1A*). Specifically, DNA methylation age as measured using the multi-tissue epigenetic clock (*Horvath, 2013*) was substantially reduced relatively early in the reprogramming process (which takes about 50 days to complete in this system), with an approximate rejuvenation of 20 years by day 10 and 40 years by day 17 (*Figure 1A*). Similar results were obtained using the skin and blood clock (*Horvath et al., 2018*;

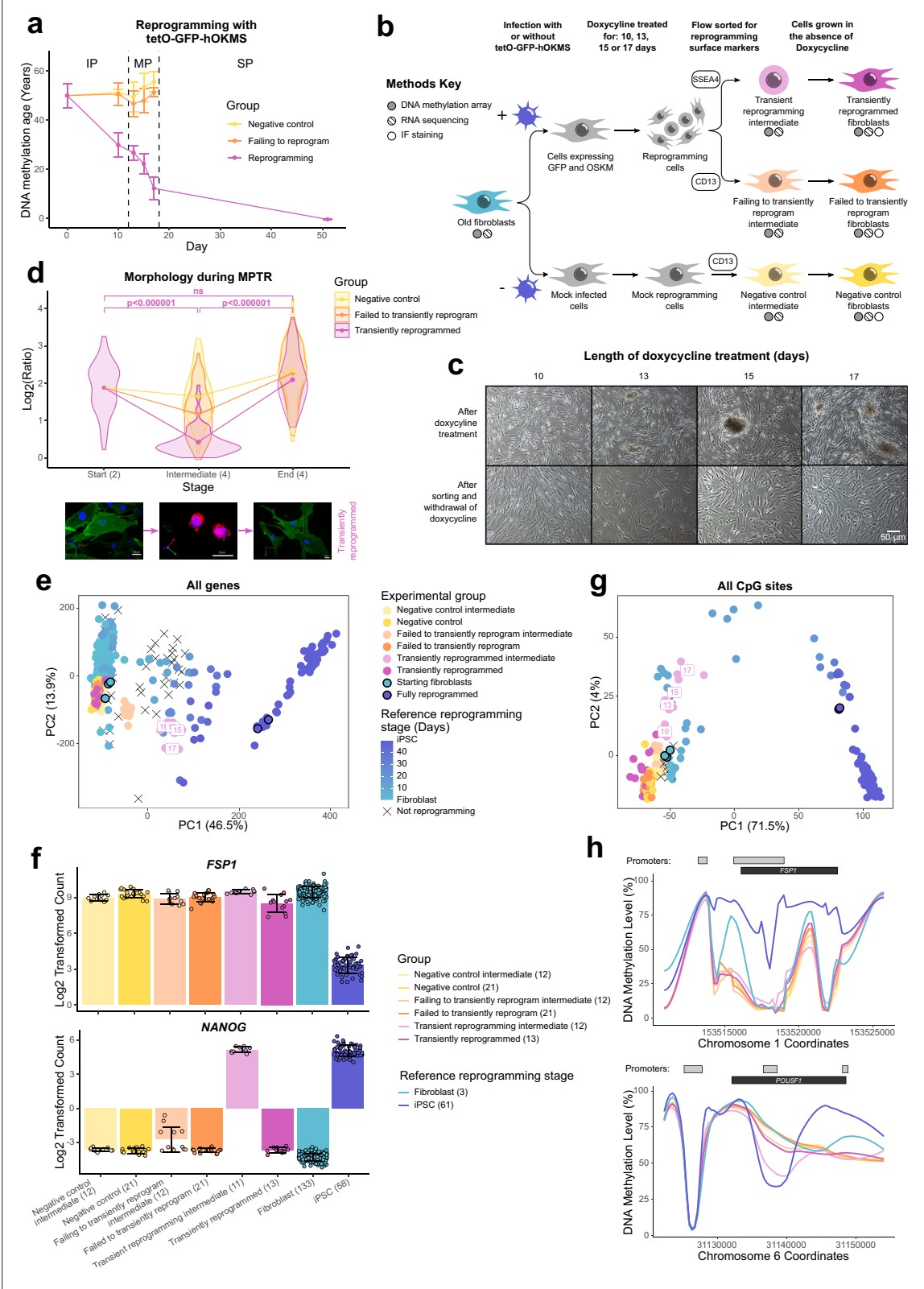

**Figure 1.** Transiently reprogrammed cells reacquire their initial cellular identity. (**A**) Mean DNA methylation age (calculated using the multi-tissue clock; *Horvath, 2013*) throughout the reprogramming process where cells were transduced with our tetO-GFP-hOKMS vector and treated continuously with 2 µg/ml of doxycycline. Reprogramming is divided in three distinct phases: initiation phase (IP), maturation phase (MP), and stabilization phase (SP). DNA methylation age decreased substantially during the MP of reprogramming in cells that were successfully reprogramming (magenta line) but not in

*Figure 1 continued*

control cells (yellow and orange lines represent non-transduced cells and cells expressing hOKMS but failing to reprogram as indicated by cell surface markers, respectively). Points represent the mean and error bars the standard deviation. N=3 biological replicates per condition, where fibroblasts were derived from different donors. N=2 biological replicates for the iPSC time point (day 51). (**B**) Experimental scheme for maturation phase transient reprogramming (MPTR). The tetO-GFP-hOKMS reprogramming construct was introduced into fibroblasts from older donors by lentiviral transduction. Alternatively, cells were 'mock infected' as a negative control. Following this, cells were grown in the presence of 2 µg/ml doxycycline to initiate reprogramming. At several time points during the MP, cells were flow sorted and successfully reprogramming cells (CD13− SSEA4+) and cells that were failing to reprogram (CD13+ SSEA4−) were collected for analysis. These were termed 'transient reprogramming intermediate' and 'failing to transiently reprogram intermediate,' respectively. Sorted cells were also further cultured, and grown in the absence of doxycycline for at least 4 weeks—these were termed 'transiently reprogrammed' (CD13− SSEA4+) or 'failed to transiently reprogram' (CD13+ SSEA4−). (**C**) Phase-contrast microscope images of cells after doxycycline treatment (transient reprogramming intermediate) and after withdrawal of doxycycline (transiently reprogrammed) as described in (**B**). The morphology of some cells changed after doxycycline treatment. These cells appeared to form colonies, which became larger with longer exposure to doxycycline. After sorting, these cells were cultured in medium no longer containing doxycycline, and appeared to return to their initial fibroblast morphology. (**D**) Roundness ratio of cells before, during, and after MPTR (with 13 days of reprogramming). Roundness ratio was calculated by dividing maximum length by perpendicular width. Fibroblasts became significantly rounder during MPTR and returned to a more elongated state upon the completion of MPTR. Values from individual cells have been represented as violin plots. Points represent mean values and are connected with lines. Significance was calculated with a Tukey's range test. Representative 3D renderings of cells (generated using Volocity) before, during, and after successful transient reprogramming are included below the plot. CD13 is colored in green, SSEA4 is colored in red, and DAPI staining is colored in blue. White scale bars represent a distance of 20 µm. (**E**) Principal component analysis of transient reprogramming and reference reprogramming sample transcriptomes (light blue to dark blue and black crosses, data from *Banovich et al., 2018*, *Fleischer et al., 2018* and our novel Sendai reprogramming data set). Reference samples form a reprogramming trajectory along PC1. In the Sendai reprogramming reference data set, cells that were not reprogramming (CD13+ SSEA4−) were also profiled and clustered midway along PC1 suggesting some transcriptional changes had still occurred in these cells. Transient reprogramming samples moved along this trajectory with continued exposure to doxycycline (light magenta points) and returned to the beginning of the trajectory after withdrawal of doxycycline (magenta points). Control samples (yellow and orange points) remained at the beginning of the trajectory throughout the experiment. (**F**) Mean gene expression levels for the fibroblast specific gene *FSP1* and the iPSC specific gene *NANOG*. Transiently reprogrammed samples expressed these genes at levels similar to control fibroblasts. Bars represent the mean and error bars the standard deviation. Samples transiently reprogrammed for 10, 13, 15, or 17 days were pooled. The number of distinct samples in each group is indicated in brackets. (**G**) Principal component analysis of transient reprogramming (magenta points) and reference reprogramming sample methylomes (light blue to dark blue and black crosses, data from *Banovich et al., 2018*, *Ohnuki et al., 2014* and our novel Sendai reprogramming data set). Reference samples formed a reprogramming trajectory along PC1. Transient reprogramming samples moved along this trajectory with continued exposure to doxycycline (light magenta points) and returned to the beginning of the trajectory after withdrawal of doxycycline (magenta points). Control samples (yellow and orange points) remained at the beginning of the trajectory throughout the experiment. (**H**) Mean DNA methylation levels across the fibroblast-specific gene *FSP1* and the iPSC-specific gene *POU5F1* (encoding OCT4). Transiently reprogrammed samples had methylation profiles across these genes that resemble those found in fibroblasts. Gray bars and black bars indicate the locations of Ensembl annotated promoters and genes, respectively. Samples transiently reprogrammed for 10, 13, 15, or 17 days were pooled for visualization purposes. The number of distinct samples in each group is indicated in brackets. iPSC, induced pluripotent stem cell; MPTR, maturation phase transient reprogramming.

The online version of this article includes the following figure supplement(s) for figure 1:

**Figure supplement 1.** Transiently reprogrammed cells reacquire their initial cellular identity.

*Figure 1—figure supplement 1A*). Interestingly, other epigenetic clocks were rejuvenated later in the reprogramming process. This may suggest that the epigenome is rejuvenated in stages; however, we note that these other epigenetic clocks were not trained on fibroblast data. We therefore focussed on the window between days 10 and 17 to develop our MPTR protocol for human fibroblasts (*Figure 1B*), predicting that this would enable substantial reversal of aging phenotypes whilst potentially allowing cells to regain original cell identity. Beyond this window, cells would enter the SP and the endogenous pluripotency genes would become activated, preventing the cessation of reprogramming by withdrawing doxycycline alone (*Samavarchi-Tehrani et al., 2010*). The reprogramming cassette was introduced into fibroblasts from three middle-aged donors (chronologically aged 38, 53, and 53 years old and epigenetically aged 45, 49, and 55 years old, according to the multi-tissue epigenetic clock *Horvath, 2013*) by lentiviral transduction before selecting successfully transduced cells by sorting for GFP. We then reprogrammed the fibroblasts for different lengths of time (10, 13, 15, or 17 days) by supplementing the media with 2 µg/ml doxycycline and carried out flow sorting to isolate cells that were successfully reprogramming (labeled 'transient reprogramming intermediate': SSEA4 positive, CD13 negative) as well as the cells that had failed to reprogram (labeled 'failing to transiently reprogram intermediate': CD13 positive, SSEA4 negative). At this stage, approximately 25% of the cells were successfully reprogramming and approximately 35% of the cells were failing to reprogram, whilst the remainder were double positive or double negative (*Figure 1—figure supplement 1B*).

Cells were harvested for DNA methylation array or RNA-seq analysis and also replated for further culture in the absence of doxycycline to stop the expression of the reprogramming cassette. Further culture for a period of 4–5 weeks in the absence of doxycycline generated 'transiently reprogrammed fibroblasts,' which had previously expressed SSEA4 at the intermediate stage, as well as 'failed to transiently reprogram fibroblasts,' which had expressed the reprogramming cassette (GFP-positive cells) but failed to express SSEA4. As a negative control, we simultaneously 'mock infected' (subject to transduction process but without lentiviruses) populations of fibroblasts from the same donors. These cells underwent an initial flow sort for viability (to account for the effects of the GFP sort) before culture under the same conditions as the reprogramming cells and flow sorting for CD13 (cells harvested at this stage generated a 'negative control intermediate' for methylome and transcriptome analyses). Finally, these 'negative control intermediate' cells were grown in the absence of doxycycline for the same length of time as experimental samples to account for the effects of extended cell culture, generating 'negative control fibroblasts' (*Figure 1B*).

After reprogramming for 10–17 days, we found the fibroblasts had undergone dramatic changes in morphology. Upon visual inspection using a light microscope it appeared that the cells had undergone a mesenchymal-to-epithelial like transition and were forming colony structures that progressively became larger with longer periods of reprogramming, consistent with the emergence of the early pluripotency marker SSEA4. After sorting the cells and culturing in the absence of doxycycline, we found they were able to return to their initial fibroblast morphology, showing that morphological reversion is possible even after 17 days of reprogramming (*Figure 1C*). We quantified the morphology changes by calculating a ratio indicative of 'roundness' (maximum length divided by perpendicular width) for individual cells before, during, and after MPTR (*Figure 1D* and *Figure 1—figure supplement 1C*). We found that successfully reprogramming cells became significantly rounder at the intermediate stages of MPTR compared to the starting fibroblasts and then returned to an elongated state upon the completion of MPTR. Of note, we found that there was no significant difference in roundness between cells before and after MPTR, further supporting that fibroblasts were able to return to their original morphology. In comparison, failing to reprogram and negative control cells did not undergo as substantial a change during MPTR and were significantly more elongated at the intermediate stage (*Supplementary file 2*).

We investigated further the identity of the cells after MPTR by conducting DNA methylation array analysis and RNA sequencing to examine their methylomes and transcriptomes, respectively. We included published reprogramming data sets in our analysis as well as a novel reprogramming data set that we generated based on Sendai virus delivery of the Yamanaka factors to act as a reference (*Fleischer et al., 2018*; *Ohnuki et al., 2014*; *Banovich et al., 2018*). Principal component analysis (PCA) using expression values of all genes in the transcriptome separated cells based on the extent of reprogramming and the reference data sets formed a reprogramming trajectory along PC1 (*Figure 1E*). Transient reprogramming intermediate cells (collected after the reprogramming phase but before the reversion phase) clustered halfway along this trajectory, implying that cells lose aspects of the fibroblast transcriptional program and/or gain aspects of the pluripotency transcriptional program, which is consistent with the loss of the fibroblast surface marker CD13 and gain of the iPSC surface marker SSEA4. We note that the different time points for the transient reprogramming intermediate samples clustered closer together when examining their transcriptomes compared to their DNA methylomes. This suggests that changes in the DNA methylome occur more gradually, whereas changes in the transcriptome occur in more discrete stages. Notably, upon completion of MPTR, transiently reprogrammed samples clustered at the beginning of this trajectory showing that these samples once again transcriptionally resemble fibroblasts rather than reprogramming intermediates or iPSCs (*Figure 1E*). Similar findings were made when the reference data sets were excluded (*Figure 1—figure supplement 1D*). For example, transiently reprogrammed cells did not express the pluripotency marker *NANOG* and expressed high levels of the fibroblast marker *FSP1* (*Figure 1F*). Notably, *NANOG* was temporarily expressed at high levels at the intermediate stages of transient reprogramming alongside FSP1, suggesting that these cells simultaneously possessed some transcriptional attributes of both fibroblasts and iPSCs.

Similarly, PCA of the methylomes separated cells based on the extent of reprogramming and the reference data sets formed a reprogramming trajectory along PC1. PC2 separated mid reprogramming samples from initial fibroblasts and final iPSCs and was driven by CpG sites that are temporarily

hypermethylated or hypomethylated during reprogramming. These CpG sites appeared near genes associated with asymmetric protein localization according to gene ontology analysis. As with the transcriptome, intermediate samples from our transient reprogramming experiment clustered along this reprogramming trajectory (*Figure 1G*), showing that cells move epigenetically toward pluripotency. Notably, the transiently reprogrammed samples returned back to the start of this trajectory (with the reference fibroblast samples) revealing that they epigenetically resembled fibroblasts once again. Like the transcriptome, similar findings were made when the reference data sets were excluded (*Figure 1—figure supplement 1E*). We found typical regions that change during reprogramming were fibroblast-like after transient reprogramming (*Takahashi et al., 2007*), such as the promoter of *POU5F1* being hypermethylated and the promoter of *FSP1* being hypomethylated in our transiently reprogrammed cells (*Figure 1H*). Notably, the *POU5F1* promoter was temporarily demethylated and the *FSP1* promoter remained lowly methylated at the intermediate stages of transient reprogramming, suggesting that these intermediate stage cells possess some epigenetic features of both fibroblasts and iPSCs. Taken together, these data demonstrate that fibroblasts can be transiently reprogrammed to the MP and then revert to a state that is morphologically, epigenetically, and transcriptionally similar to the starting cell identity. To our knowledge, this is the first method for MPTR, where Yamanaka factors are transiently expressed up to the MP of reprogramming before the expression of the factors is abolished.

## Epigenetic memory and transcriptional persistence are present at the intermediate stages of transient reprogramming

Though transiently reprogrammed fibroblasts temporarily lost their cell identity (becoming SSEA4 positive and CD13 negative), they were able to reacquire it once the reprogramming factors were removed, suggesting that they retained memory of their initial cell identity. To examine the source of this memory, we initially defined fibroblast-specific and iPSC-specific gene sets using differential expression analysis on fibroblasts before and after complete reprogramming with our system (*Figure 2—figure supplement 1A*). We subsequently analyzed the expression of these gene sets throughout MPTR and observed that fibroblast-specific genes were temporarily downregulated whilst iPSC-specific genes were temporarily upregulated (*Figure 2A*). As expected, these gene sets were further downregulated and upregulated during complete reprogramming, respectively (*Figure 2A*). We note that this approach generalizes the expression changes and as a result, may obscure subclusters within these gene sets that display different expression trajectories. Therefore, we analyzed the expression levels of individual genes to gain further insight into these gene sets. After performing hierarchical clustering, we observed that the majority of genes within the fibroblast-specific gene set were temporarily downregulated during transient reprogramming (2803 genes out of 4178). However, we also observed that the remaining genes formed two additional clusters that were temporarily upregulated (961 genes) and persistently expressed (414 genes), respectively (*Figure 2B*, *Figure 2—figure supplement 1B* and *Supplementary file 3*). We also clustered the genes within the iPSC-specific gene set and observed that the majority of iPSC genes were upregulated in transient reprogramming intermediate cells to levels similar to iPSCs and the remaining genes were not yet activated (*Figure 2—figure supplement 1C*). We subsequently performed gene ontology analysis on the fibroblast-specific gene clusters and found that the temporarily upregulated cluster was enriched for gene ontology categories such as 'response to lipopolysaccharide' suggesting that inflammatory signaling pathways are temporarily activated during transient reprogramming, likely in response to the reprogramming factors. Interestingly, the persistently expressed gene cluster was enriched for gene ontology categories such as extracellular matrix and collagen fibril organization, suggesting that some aspects of fibroblast function are maintained during transient reprogramming at least at the transcriptional level (*Figure 2—figure supplement 1D*).

We also questioned whether the epigenome played a role in the retention of memory of the initial cell type, particularly for genes that were temporarily downregulated. We therefore examined the DNA methylation levels at regulatory elements linked to the fibroblast-specific genes. We used the Ensembl Regulatory Build (*Zerbino et al., 2015*) to obtain the locations of promoter and enhancer elements as well as their activity status in dermal fibroblasts and iPSCs. We then focussed on promoter and enhancer elements that are active in fibroblasts and linked them to the nearest transcription start site (within 1 kb for promoters and 1 mb for enhancers). The promoters associated with fibroblast

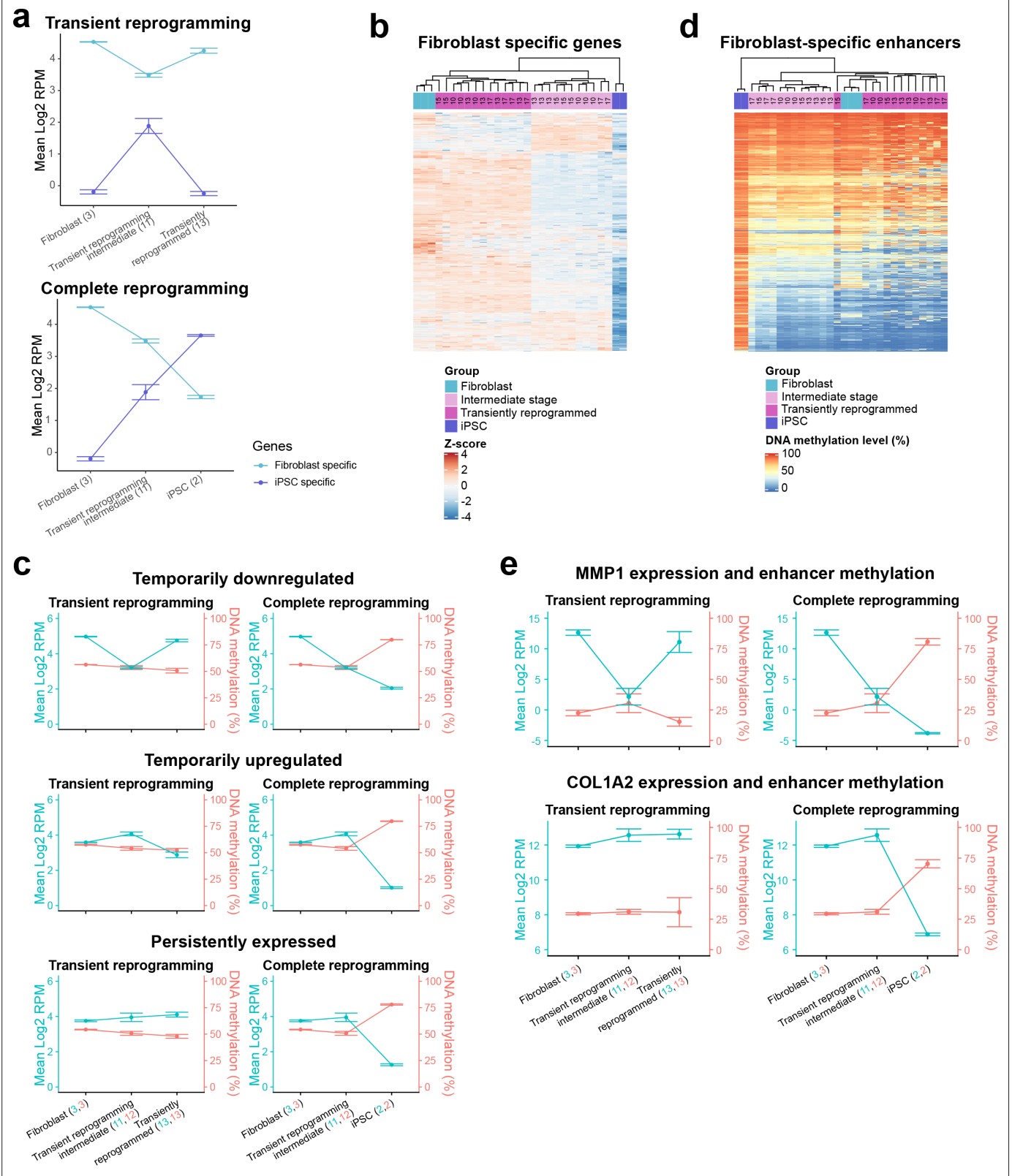

**Figure 2.** Epigenetic memory at enhancers and persistent fibroblast gene expression may allow cells to return to their initial identity. (**A**) The mean expression levels of fibroblast-specific and iPSC-specific gene sets during transient reprogramming and complete reprogramming. Error bars represent the standard deviation. (**B**) Heatmap examining the expression of fibroblast-specific genes in cells before (light blue group), during (light magenta group, transient reprogramming intermediate cells), and after (magenta group, transiently reprogrammed fibroblasts) transient reprogramming as

*Figure 2 continued on next page*

*Figure 2 continued*

well as in iPSCs (dark blue group). The number of days of reprogramming is indicated above the heatmap where applicable. The majority of fibroblast genes are downregulated at the intermediate stages of transient reprogramming. However, some fibroblast genes are persistently expressed or temporarily upregulated at this stage. (**C**) Mean DNA methylation levels across enhancers linked to the three clusters of fibroblast genes during transient reprogramming and complete reprogramming. DNA methylation levels across enhancers remain unchanged during transient reprogramming regardless of the expression of their associated genes. In comparison, DNA methylation levels across these regions increase during complete reprogramming. Error bars represent the standard deviation. (**D**) Heatmap examining the DNA methylation levels of fibroblast-specific enhancers in cells before (light blue group), during (light magenta group), and after (magenta group) transient reprogramming as well as in iPSCs (dark blue group). Each sample was plotted as a single column, whether reprogrammed for 10, 13, 15, or 17 days. Fibroblast enhancers became hypermethylated during complete reprogramming but were still demethylated at the intermediate stages of transient reprogramming. Fibroblast-specific enhancers were defined as enhancers that are active in fibroblasts but no longer active in iPSCs (become inactive, poised, or repressed) based on Ensembl regulatory build annotations. (**E**) The mean expression and enhancer methylation levels of example genes during transient reprogramming and complete reprogramming. MMP1 is a gene that demonstrates epigenetic memory as it is temporarily downregulated during transient reprogramming and its enhancer remains demethylated. COL1A2 is a gene that demonstrates transcriptional persistence as it remains expressed throughout transient reprogramming. iPSC, induced pluripotent stem cell.

The online version of this article includes the following figure supplement(s) for figure 2:

**Figure supplement 1.** Epigenetic memory at enhancers and persistent fibroblast gene expression may allow cells to return to their initial identity.

genes remained lowly methylated throughout transient reprogramming and complete reprogramming regardless of the gene cluster, suggesting that promoter methylation does not contribute substantially toward memory (*Figure 2—figure supplement 1E*). In contrast, enhancers associated with fibroblast genes gained DNA methylation but only during complete reprogramming and not during transient reprogramming (*Figure 2C* and *Figure 2—figure supplement 1F*). This was the case for enhancers linked to the genes in all three clusters and in the case of temporarily downregulated genes, the lack of hypermethylation may confer epigenetic memory at a time when the associated genes are transcriptionally repressed. We also examined fibroblast-specific enhancers in general and defined these as enhancers that are active in fibroblasts but are no longer active in iPSCs. Similar to the previous analysis, we found that DNA methylation was relatively dynamic at fibroblast-specific enhancers. Approximately half of all fibroblast-specific enhancers (2351 out of the covered 4204 enhancers) gained DNA methylation during iPSC reprogramming. However, even at day 17 of the reprogramming process (the longest transient reprogramming intermediate tested here), these enhancers still remained hypomethylated (*Figure 2D*). Overall, we hypothesize that both epigenetic memory at genes such as MMP1 (*Figure 2E*) and transcriptional persistence at genes such as COL1A2 (*Figure 2E*) enable cells to return to their original cell type once the reprogramming factors are withdrawn. Taken together, these two attributes may act as the source of memory for initial cell identity during a time when the somatic transcriptional program is otherwise mostly repressed and somatic proteins such as CD13 are lost (*Polo et al., 2012*; *David and Polo, 2014*).

## Transient reprogramming reverses age-associated changes in the transcriptome and partially restores fibroblast function

We next investigated the transcriptome to determine if there was any evidence of rejuvenation in this omic layer. We initially identified genes that significantly correlated with age in a reference fibroblast aging data set (*Fleischer et al., 2018*) and used genes with a significant Pearson correlation after Bonferroni correction ($p \leq 0.05$) to carry out PCA (3707 genes). The samples primarily separated by age and reference fibroblast samples formed an aging trajectory. The transiently reprogrammed samples clustered closer to the young fibroblasts along PC1 than the negative control samples (*Figure 3A*). Based on the relationship between PC1 and age in the reference data set, we inferred that transient reprogrammed samples were approximately 40 years younger than the negative control samples (*Figure 3B*). To further quantify the extent of rejuvenation, we investigated the effect of MPTR using transcription clocks. Unfortunately, existing transcription clocks failed to accurately predict the age of our negative control samples. This may be due to batch effects such as differences in RNA-seq library preparation and data processing pipelines. To overcome this problem, we trained a transcription age-predictor using random forest regression on published fibroblast RNA-seq data from donors aged 1–94 years old that was batch corrected to our transient reprogramming data set (*Fleischer et al., 2018*). The transcription age predictor was trained on transformed age, similar to the Horvath

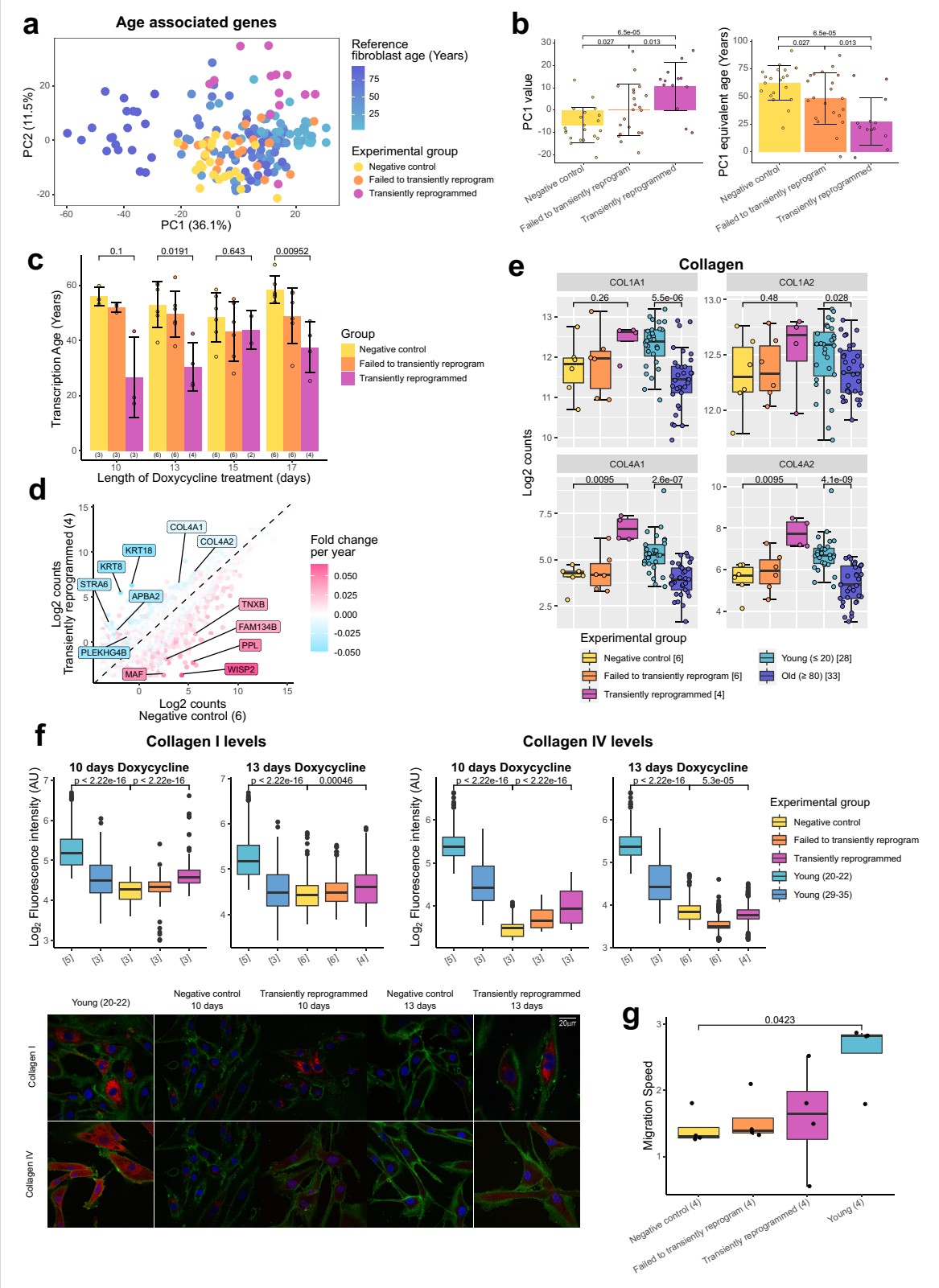

**Figure 3.** Transient reprogramming reverses age-associated changes in the transcriptome and partially restores fibroblast migration speed. (**A**) Principal component analysis (PCA) of fibroblast aging-associated gene expression levels in transient reprogramming (magenta) and reference aging fibroblast samples (light blue-dark blue). Reference samples formed an aging trajectory along PC1. Transiently reprogrammed samples located closer to young fibroblasts than negative control samples did (yellow and orange), suggesting they were transcriptionally younger. (**B**) PC1 values from the PCA of

*Figure 3 continued on next page*

*Figure 3 continued*

fibroblast aging-associated gene expression levels and their equivalent age based on the reference aging fibroblast samples. PC1 values were greater in transiently reprogrammed samples than negative control and failed to transiently reprogram samples and as result these samples appear to be younger. Bars represent the mean and error bars represent the standard deviation. (**C**) Mean transcription age calculated using a custom transcriptome clock (median absolute error = 12.57 years) for negative control samples (yellow), samples that expressed OSKM but failed to reprogram based on cell surface markers (orange) and cells that were successfully transiently reprogrammed (magenta) as described in *Figure 1B* for 10, 13, 15, or 17 days. The number of distinct samples in each group is indicated in brackets. Bars represent the mean and error bars the standard deviation. Statistical significance was calculated with Mann-Whitney U-tests. (**D**) The mean expression levels of all genes in transiently reprogrammed samples with 13 days of reprogramming compared to those in corresponding negative control samples. In addition, genes have been color coded by their expression change with age. Genes that upregulate with age were downregulated with transient reprogramming and genes that downregulate with age were upregulated with transient reprogramming. Notable example genes have been highlighted. The number of distinct samples in each group is indicated in brackets. (**E**) The expression levels of collagen genes that were restored to youthful levels after transient reprogramming with 13 days of reprogramming. Bars represent the mean and error bars the standard deviation. The number of distinct samples in each group is indicated in square brackets. Significance was calculated with a two-sided Mann-Whitney U-test. (**F**) Boxplots of the protein levels of collagen I and IV in individual cells after transient reprogramming for 10 or 13 days calculated based on fluorescence intensity within segmented cells following immunofluorescence staining. Boxes represent upper and lower quartiles and central lines the median. The protein levels of collagen I and IV increased after transient reprogramming. The number of distinct samples in each group is indicated in square brackets. Representative images are included (bottom panel). CD44 is colored in green, collagen I and IV are colored in red, and DAPI staining is colored in blue. Significance was calculated with a two-sided Mann-Whitney U-test. (**G**) The migration speed of fibroblasts in a wound healing assay. Migration speed was significantly lower in negative control fibroblasts from middle-aged donors compared to fibroblasts from young donors (aged 20–22). Transient reprogramming improved the migration speed in some samples but had no effect in others. Technical replicates were averaged, and the mean values have been presented as boxplots where the boxes represent the upper and lower quartiles and the central lines the median. Significance was calculated with a Tukey's range test.

The online version of this article includes the following figure supplement(s) for figure 3:

**Figure supplement 1.** Transient reprogramming reverses age-associated changes in the transcriptome and partially restores fibroblast migration speed.

epigenetic clock, to account for the accelerated aging rate during childhood and adolescence (*Horvath, 2013*). The final transcription age predictor had a median absolute error of 12.57 years (*Figure 3—figure supplement 1A*), this error being higher than that of the epigenetic clock consistent with previous transcription age predictors (*Peters et al., 2015*; *Fleischer et al., 2018*). Using our predictor, we found that transient reprogramming reduced mean transcription age by approximately 30 years (*Figure 3C*). We also observed a moderate reduction in transcription age in cells that failed to transiently reprogram (SSEA4 negative at the intermediate time point), suggesting expression of the reprogramming factors alone was capable of rejuvenating some aspects of the transcriptome. Interestingly, we observed that MPTR with longer reprogramming phases reduced the extent of rejuvenation, suggesting that 10 or 13 days may be the optimum for transcriptional rejuvenation. We note that the reduction in transcription age from MPTR appears to be greater than that recently achieved by transient transfection of the Yamanaka factors (*Sarkar et al., 2020*), which was by approximately 10 years according to our transcription age predictor (*Figure 3—figure supplement 1B*), consistent with our approach of reprogramming further into the MP rather than only up to the end of the IP. Recently, a novel transcription clock called BiT age clock has been defined (*Meyer and Schumacher, 2021*), which has been trained on binarized gene expression levels. This clock has a very low median absolute error, which is comparable to that of epigenetic clocks. We ran a retrained version of the BiT age clock on our data set and made similar findings to our random forest-based clock. Of note, we observed that transient reprogramming also rejuvenated the BiT age clock by approximately 20 years relative to negative controls and that 10 or 13 days of reprogramming was optimal for maximal transcriptional rejuvenation (*Figure 3—figure supplement 1C*).

We further profiled the effects of MPTR with 13 days of reprogramming (due to its apparent significance) by examining the whole transcriptome. This was achieved by comparing the expression levels of genes in transiently reprogrammed cells to those in negative control cells and subsequently overlaying the expression change due to age calculated using the reference aging data set (*Fleischer et al., 2018*). As expected, we observed an overall reversal of the aging trends, with genes upregulated during aging being downregulated following transient reprogramming and genes downregulated during aging being upregulated following transient reprogramming (*Figure 3D*, *Figure 3—figure supplement 1D*). Notably, structural proteins downregulated with age that were upregulated upon transient reprogramming included the cytokeratins 8 and 18 as well as subunits of collagen IV.

The production of collagens is a major function of fibroblasts (*Humphrey et al., 2014*), thus we examined the expression of all collagen genes during fibroblast aging and after transient reprogramming with 13 days of reprogramming (*Figure 3E*). As shown previously (*Varani et al., 2006*; *Lago and Puzzi, 2019*), we found collagen I and IV were downregulated with age, with collagen IV demonstrating a more dramatic reduction. Notably, the expression of both genes was restored to youthful levels after transient reprogramming, though this was not significant for collagen I likely due to the small expression difference associated with age and lower number of samples (*Figure 3E*). We then assessed by immunofluorescence whether this increased mRNA expression resulted in increased protein levels and indeed found that transient reprogramming resulted in an increase in collagen I and IV protein toward more youthful levels (*Figure 3F*). Fibroblasts are also involved in wound healing responses (*Li and Wang, 2011*), so we investigated the impact of transient reprogramming on this function using an in vitro wound healing assay (*Figure 3G* and *Figure 3—figure supplement 1E*). We found that migration speed was significantly reduced in our control fibroblasts from middle-aged donors compared to fibroblasts from young donors (aged 20–22 years old). Transient reprogramming improved the median migration speed, however, the individual responses were quite variable and in some cases migration speed was improved and in other cases it was unaffected. Interestingly, this did not appear to correlate with other aging measures such as transcription and methylation clocks. Our data show that transient reprogramming followed by reversion can rejuvenate fibroblasts both transcriptionally and at the protein level, at least based on collagen production, and functionally at least in part. This indicates that our rejuvenation protocol can, in principle, restore youthful functionality in human cells.

## Optimal transient reprogramming reverses age-associated changes in the epigenome

After finding evidence of transcriptomic rejuvenation, we sought to determine whether there were also aspects of rejuvenation in the epigenome. We initially examined global levels of H3K9me3 by immunofluorescence. H3K9me3 is a histone modification associated with heterochromatin that has been previously shown to be reduced globally with age in a number of organisms (*Ni et al., 2012*), including in human fibroblasts (*O'Sullivan et al., 2010*; *Scaffidi and Misteli, 2006*). We were able to confirm this observation and found that MPTR was able to substantially reverse this age-associated reduction back to a level comparable with fibroblasts from younger donors (with a mean age of 33 years old). Both 10 and 13 days of transient reprogramming increased global levels of H3K9me3 suggesting that this epigenetic mark, similar to the transcriptome, has a relatively broad window for rejuvenation by transient reprogramming. We also observed a slight increase in H3K9me3 levels in cells that failed to transiently reprogram, suggesting that expression of the reprogramming factors alone is capable of partially restoring this epigenetic mark (*Figure 4A*), as was observed for our transcriptome-based age-predictor (*Figure 3C*). The magnitude of rejuvenation in H3K9me3 levels in our transiently reprogrammed cells is similar to that observed from IP transient reprogramming (*Sarkar et al., 2020*).

We next applied the epigenetic clock, a multi-tissue age predictor that predicts age based on the DNA methylation levels at 353 CpG sites (*Horvath, 2013*), to our data. Notably, with 13 days of transient reprogramming, we observed a substantial reduction of the median DNA methylation age—by approximately 30 years, quantitatively the same rejuvenation as we saw in the transcriptome (*Figure 4B*). A shorter period of transient reprogramming (10 days) resulted in a smaller reduction of DNA methylation age, consistent with our results profiling DNA methylation age throughout the reprogramming process, where DNA methylation age gradually reduced throughout the MP (*Figure 1A*). This epigenetic rejuvenation is potentially promoted by de novo methylation and active demethylation as the de novo methyltransferases and TET enzymes are upregulated during the MP (*Figure 4—figure supplement 1A*). Potentially, some of the rejuvenating mechanisms occurring in MPTR may mirror those that occur during embryonic development as epigenetic rejuvenation during embryonic development coincides with de novo methylation of the genome (*Kerepesi et al., 2021*). Similar to the transcription clocks, we also observed a smaller reduction in DNA methylation age with longer transient reprogramming times, suggesting that some aspects of the observed epigenetic rejuvenation are lost during the reversion phase of our MPTR protocol. Potentially, extended reprogramming (for 15 or 17 days) may make reversion more difficult and result in cellular stresses that 're-age'

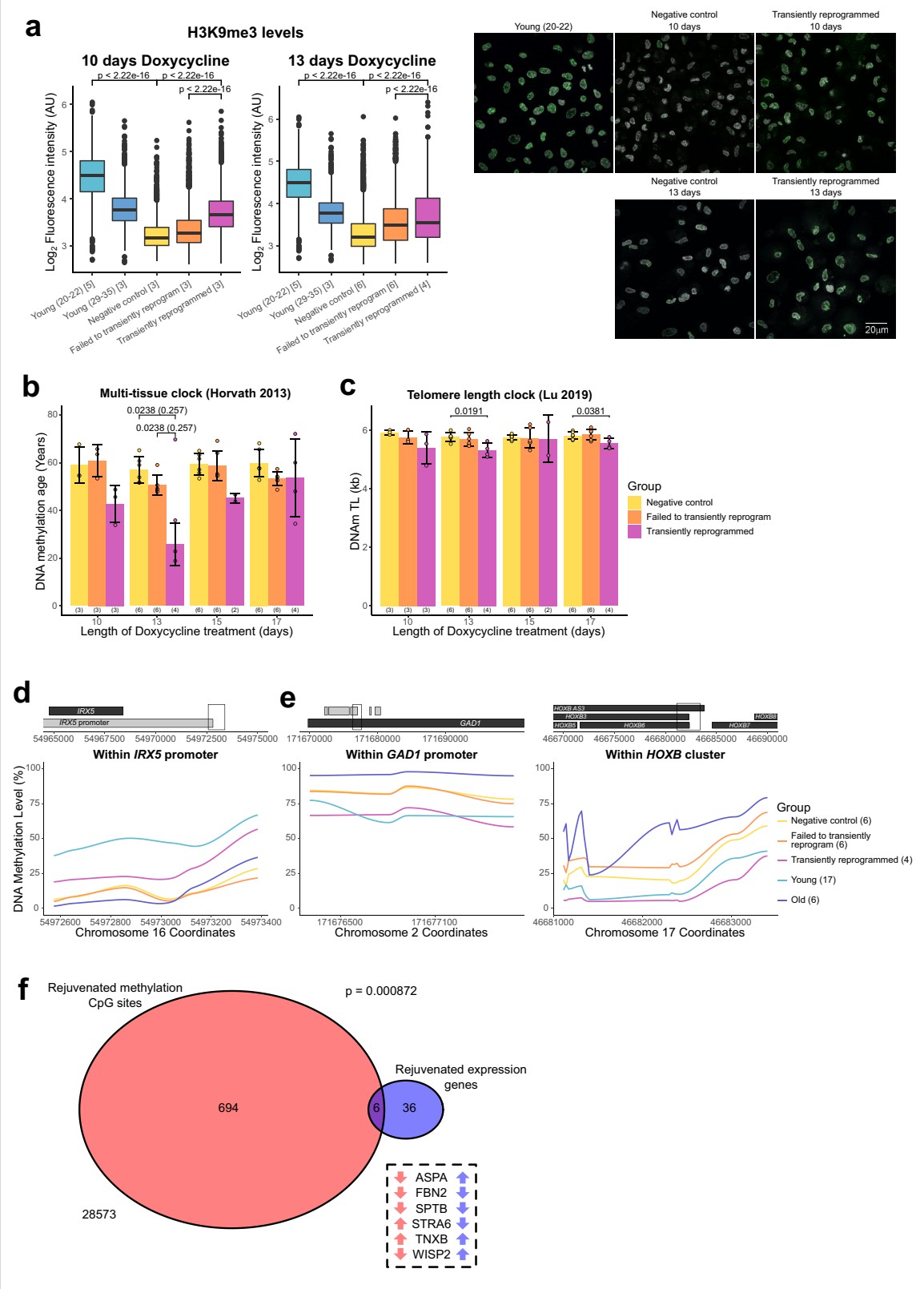

**Figure 4.** Optimal transient reprogramming can reverse age-associated changes in the epigenome. (**A**) Boxplots of the levels of H3K9me3 in individual cells calculated based on fluorescence intensity within nuclei (segmented using DAPI). The levels of H3K9me3 were found to decrease with age and increase after transient reprogramming for 10 or 13 days. Boxes represent upper and lower quartiles and central lines the median. The number of distinct samples in each group is indicated in square brackets. Representative images are included (right panel). H3K9me3 is colored in green and DAPI

*Figure 4 continued on next page*

*Figure 4 continued*

staining is colored in gray scale. Significance was calculated with a two-sided Mann-Whitney U-test. (**B**) Mean DNA methylation age of samples after transient reprogramming calculated using the multi-tissue clock (*Horvath, 2013*). DNA methylation age substantially reduced after 13 days of transient reprogramming. Shorter and longer lengths of transient reprogramming led to smaller reductions in DNA methylation age. Bars represent the mean and error bars represent the standard deviation. The outlier in the 13 days of transient reprogramming group was excluded from calculation of the mean and standard deviation. Significance was calculated with a two-sided Mann-Whitney U-test with (in brackets) and without the outlier. The number of distinct samples in each group is indicated in brackets beneath the bars. (**C**) Mean telomere length of samples after transient reprogramming calculated using the telomere length clock (*O'Sullivan et al., 2010*). Telomere length either did not change or was slightly reduced after transient reprogramming. Bars represent the mean and error bars represent the standard deviation. Significance was calculated with a two-sided Mann-Whitney U test. (**D**) Mean DNA methylation levels across a rejuvenated age-hypomethylated region. This region is found within the IRX5 promoter. Samples transiently reprogrammed for 13 days were pooled for visualization purposes. The number of distinct samples in each group is indicated in brackets. (**E**) Mean DNA methylation levels across rejuvenated age-hypermethylated regions. These regions are found within the GAD1 promoter and HOXB locus. Samples transiently reprogrammed for 13 days were pooled for visualization purposes. The number of distinct samples in each group is indicated in brackets. (**F**) The overlap in rejuvenated methylation CpG sites and rejuvenated expression genes. Rejuvenated CpG sites were annotated with the nearest gene for this overlap analysis. The universal set was restricted to genes that were annotated to CpG sites in the DNA methylation array. Fisher's exact test was used to calculate the significance of the overlap. The six genes that were found in both sets are listed along with the direction of their DNA methylation (red) and gene expression (blue) change with age.

The online version of this article includes the following figure supplement(s) for figure 4:

**Figure supplement 1.** Optimal transient reprogramming can reverse age-associated changes in the epigenome.

the methylome during the process. Similar results were obtained using the skin and blood clock and the Weidner clock (*Horvath et al., 2018*; *Weidner et al., 2014*; *Figure 4—figure supplement 1B*). Other epigenetic clocks were not rejuvenated by MPTR; however, we note that these clocks either rejuvenate later in the reprogramming process or are unaffected by reprogramming (*Figure 1—figure supplement 1A*).

Telomeres are protective structures at the ends of chromosomes that consist of repetitive sequences. Telomere length decreases with age due to cell proliferation in the absence of telomerase enzymes and is restored upon complete iPSC reprogramming (*Lapasset et al., 2011*). To investigate the effect of transient reprogramming on telomere length, we used the telomere length clock, which predicts telomere length based on the methylation levels at 140 CpG sites (*Lu et al., 2019b*). We found that MPTR does not affect telomere length and, in some cases, slightly reduces it (*Figure 4C*). This is consistent with our results profiling telomere length throughout complete reprogramming using our doxycycline inducible system, where telomere length did not increase until the SP (*Figure 4—figure supplement 1C*). This coincides with the expression of telomerase during reprogramming, where it is weakly expressed during the later stages of the MP and only strongly expressed during the SP (*Figure 4—figure supplement 1D*).

Next, we investigated the locations of the rejuvenated CpG sites and found that most were individual sites spread across the genome (*Figure 4—figure supplement 1E*). Some of these individual CpG sites may be part of larger regions of rejuvenated methylation, which we are unable to fully detect due to the targeted nature of DNA methylation array profiling; however, we found a few small clusters of rejuvenated CpG sites. We found that a small region in the *IRX5* promoter became demethylated with age and transient reprogramming was able to partially remethylate this region (*Figure 4D*). IRX5 is involved in embryonic development so demethylation of its promoter with age may lead to inappropriate expression (*Costantini et al., 2005*; *Cheng et al., 2005*). We also found two regions that became hypermethylated with age and were demethylated by transient reprogramming (*Figure 4E*). One of these regions is in the *GAD1* promoter; encoding an enzyme that catalyzes the conversion of gamma-aminobutyric acid into glutamic acid (*Bu et al., 1992*). The other region is within the *HOXB* locus, involved in anterior-posterior patterning during development (*Pearson et al., 2005*). Finally, we examined whether there was any overlap between the epigenetic and transcriptional rejuvenation. We therefore annotated the rejuvenated CpG sites with the nearest gene and then overlapped this gene set with the list of genes with rejuvenated expression. We found that there was a significant overlap between these two groups suggesting that epigenetic rejuvenation and transcriptional rejuvenation may be partially linked (*Figure 4F*). We further examined these overlapping genes and found that several had structural roles. These included *FBN2* and *TNXB*, which encode components of the extracellular matrix (*Zhang et al., 1994*; *Bristow et al., 1993*) and *SPTB*, which encodes a component of the cytoskeletal network (*Garbe et al., 2007*). WISP2 was also rejuvenated transcriptionally and

epigenetically; this gene is an activator of the canonical WNT pathway (*Grünberg et al., 2014*) and has recently been shown to inhibit collagen linearisation (*Janjanam, 2021*). ASPA and STRA6 respectively encode an enzyme that hydrolyses N-acetyl-I-aspartate and a vitamin A receptor (*Bitto et al., 2007*; *Amengual et al., 2014*). Neither of these genes have obvious roles in fibroblasts. We note that additional overlaps between epigenetic and transcriptional rejuvenation may exist that are not observed in our study due to the limited genomic coverage of DNA methylation arrays. Overall, our data demonstrate that transient reprogramming for 13 days (but apparently not for longer or shorter periods) represents a 'sweet spot' that facilitates partial rejuvenation of both the methylome and transcriptome, reducing epigenetic and transcriptional age by approximately 30 years.

## Discussion

Here, we have developed a novel method, MPTR, where the Yamanaka factors are ectopically expressed until the MP of reprogramming is reached, and their induction is then withdrawn. MPTR rejuvenates multiple molecular hallmarks of aging robustly and substantially, including the transcriptome, epigenome, functional protein expression, and cell migration speed. Previous attempts at transient reprogramming have been restricted to the IP in order to conserve initial cell identity (*Ocampo et al., 2016*; *Lu et al., 2020*; *Sarkar et al., 2020*). This is a valid concern as fully reprogrammed iPSCs can be difficult to differentiate into mature adult cells and instead these differentiated cells often resemble their fetal counterparts (*Hrvatin et al., 2014*). With our approach, cells temporarily lose their identity as they enter the MP but, importantly, reacquire their initial somatic fate when the reprogramming factors are withdrawn. This may be the result of persisting epigenetic memory at enhancers (*Jadhav et al., 2019*), which notably we find is not erased until the SP, as well as persistent expression of some fibroblast genes.

With our method employing longer periods of reprogramming, we observed robust and substantial rejuvenation of the whole transcriptome as well as aspects of the epigenome, with many features becoming approximately 30 years younger. This extent of rejuvenation appears to be substantially greater than what has been observed previously for transient reprogramming approaches that reprogram within the IP. The methylome appears to require longer reprogramming to substantially rejuvenate and consequently, previous work using shorter lengths of reprogramming resulted in modest amounts of rejuvenation of the methylome (*Lu et al., 2020*; *Sarkar et al., 2020*). However, we note that future studies are required to thoroughly compare these approaches with our method, ideally being performed in parallel on the same starting material and with the same reprogramming system, especially as different reprogramming systems can reprogram cells at different speeds (*Schlaeger et al., 2015*). Interestingly, these findings demonstrate that different parts of the epigenome undergo contrasting changes during transient reprogramming with age-associated CpG sites becoming differentially methylated during the MP and cell-identity regions remaining unchanged until the SP. The CpG sites within these two categories are distinct and the differential timing may suggest that different and potentially specific mechanisms are responsible for these changes. Telomere attrition is another aging hallmark, which can induce DNA damage and senescence (*López-Otín et al., 2013*). Consistent with previous studies (*Marion et al., 2009*), our reprogramming system did not induce telomere elongation until the SP, likely explaining why telomeres were not elongated by MPTR.

More recently, there have been in vivo transient reprogramming approaches that elicit similar magnitudes of rejuvenation to our in vitro MPTR method. In mice, 1 week of reprogramming induction followed by 2 weeks of recovery reversed age-associated expression changes (including collagen gene expression) and partially rejuvenated the DNA methylome in the pancreas (*Chondronasiou et al., 2022*). Interestingly, these outcomes closely mirror those observed in our human fibroblasts after MPTR. We note that iPSC reprogramming proceeds faster in mouse cells than in human cells (*Teshigawara et al., 2017*) and so this in vivo approach likely also reprograms up to the MP, supporting our findings that transient reprogramming up to the MP can substantially reverse multiple features of aging. In another recent approach, reprogramming was cyclically induced in mice for 2 days followed by 5 days of recovery for 7 months. This substantially reversed epigenetic clocks by up to 0.4 years (equivalent to 20 years in humans, similar to our system) (*Browder, 2022*). These results suggest that the rejuvenation from shorter periods of transient reprogramming is additive and when performed long term can reach the magnitude elicited by MPTR.

Quantifying the age of the transcriptome is challenging and our attempts to quantify transcriptional rejuvenation suggested varying magnitudes ranging from 20 to 40 years. In addition, we needed to apply batch correction to compare to reference aging data sets. There is a need in the field for a more robust transcription clock that can predict age accurately and can be applied to other data sets without the need to batch correct. Such a tool would be invaluable and enable us to quantify more accurately the true extent of transcriptional rejuvenation arising from MPTR.

Upon further interrogation of the transcriptomic rejuvenation, we also observed changes in genes with non-fibroblast functions. In particular, the age-associated downregulation of *APBA2* and the age-associated upregulation of *MAF* were reversed (*Figure 3D*). APBA2 stabilizes amyloid precursor protein, which plays a key role in the development of Alzheimer's disease (*Araki, 2003*). MAF regulates the development of embryonic lens fibre cells, and defects in this gene lead to the development of cataracts, which are a frequent complication in older age (*Ring et al., 2000*). These observations may signal the potential of MPTR to promote more general rejuvenation signatures that could be relevant for other cell types such as neurons. It will be interesting to determine if MPTR-induced rejuvenation is possible in other cell types, which could help us understand and potentially treat age-related diseases such as Alzheimer's disease and cataracts. Potentially we may be able to rejuvenate ex vivo clinically relevant cell types and administer these rejuvenated cells as an autologous cell therapy, for example, fibroblasts rejuvenated by MPTR may be applicable for treating skin wounds and improving wound healing. In addition, we may be able to use MPTR as a screening platform to find novel candidate genes that are responsible for reversing age-associated changes during reprogramming. Potentially by targeting such genes, we may be able to reverse age-associated changes without inducing pluripotency.

In our study, we investigated different lengths of reprogramming for our MPTR method and surprisingly found that longer lengths of reprogramming did not always promote more rejuvenation in the transcriptome and epigenome. Instead, we found that 13 days of reprogramming was the optimal period and that longer lengths of reprogramming diminished the extent of transcriptional and epigenetic rejuvenation. This finding contrasts with the observations of cells undergoing complete iPSC reprogramming and highlights the importance of assessing multiple reprogramming durations when using transient reprogramming approaches.

The Yamanaka factors possess oncogenic properties, which can lead to teratoma formation when persistently overexpressed in vivo (*Abad et al., 2013*; *Ohnishi et al., 2014*). Our approach should avoid these properties as we only temporarily express the factors, similar to other transient reprogramming approaches (*Ocampo et al., 2016*; *Sarkar et al., 2020*). Whilst we could not find any signatures of pluripotency within the transcriptomes or methylomes of transiently reprogrammed cells, we cannot discount the possibility that a minor subset of cells within the population maintain pluripotent-like characteristics, and could therefore induce teratoma formation if transplanted in vivo. We note though that this is a proof-of-concept study and that the method will eventually require modifications to be more suitable for therapeutic applications, such as by replacing the lentiviral vectors with non-integrating vectors.

The effect of starting age is a factor that remains to be explored. In our study, we examined the effects of MPTR on fibroblasts from middle-aged donors and observed an approximately 30-year rejuvenation. It will be interesting to perform our method on fibroblasts from younger and older donors to see if the rejuvenating effect of MPTR is constant. In that case, cells would always become 30 years younger than their controls. Alternatively, the effect of MPTR may scale with starting age, with more rejuvenation being observed in cells from older donors compared to cells from younger donors. Finally, we note that multiple cycles of transient reprogramming can be performed with some approaches (*Ocampo et al., 2016*). It will be interesting to examine if MPTR can be performed repeatedly on cells and if this may improve the extent of rejuvenation. However, this may not be possible with our current system as telomere length is unaffected by MPTR. In addition, multiple cycles may not improve the extent of rejuvenation as there may be a minimum age that can be achieved when limiting reprogramming to the MP.

Overall, our results demonstrate that substantial rejuvenation is possible without acquiring stable pluripotency and suggest the exciting concept that the rejuvenation program may be separable from the pluripotency program. Future studies are warranted to determine the extent to which these two programs can be separated and could lead to the discovery of novel targets that promote rejuvenation without the need for iPSC reprogramming.

## Materials and methods

### Plasmids and lentivirus production

The doxycycline-inducible polycistronic reprogramming vector was generated by cloning a GFP-IRES sequence downstream of the tetracycline response element in the backbone FUW-tetO-hOKMS (Addgene 51543, a gift from *Cacchiarelli et al., 2015*). This vector was used in combination with FUW-M2rtTA (Addgene 20342, a gift from *Hockemeyer et al., 2008*). Viral particles were generated by transfecting HEK293T cells with the packaging plasmids pMD2.G (Addgene 12259, a gift from Didier Trono) and psPAX2 (Addgene 12260, a gift from Didier Trono) and either FUW-tetO-GFP-hOKMS or FUW-M2rtTA.

### iPSC reprogramming

Dermal fibroblasts from middle-aged donors (38–53 years old) were purchased from Lonza and Gibco and were used at passage 4 after purchase for reprogramming experiments. Cells were routinely tested for mycoplasma. For lentiviral iPSC reprogramming, fibroblasts were expanded in fibroblast medium (Dulbecco's modified Eagle's medium [DMEM]-F12, 10% fetal bovine serum [FBS], 1× Glutamax, 1× MEM-NEAA, 1× beta-mercaptoethanol, 0.2× penicillin/streptomycin, and 16 ng/ml FGF2) before being spinfected with tetO-GFP-hOKMS and M2rtTA lentiviruses, where 10% virus supernatant and 8 µg/ml polybrene was added to the cells before centrifugation at 1000 rpm for 60 min at 32°C. Reprogramming was initiated 24 hr after lentiviral transduction by introducing doxycycline (2 µg/ml) to the media. Media were changed daily throughout the experiment subsequently. On day 2 of reprogramming, cells were flow sorted for viable GFP positive cells and then cultured on gelatine coated plates. On day 7 of reprogramming, cells were replated onto irradiated mouse embryonic fibroblasts (iMEFs) and on day 8 of reprogramming, the medium was switched to hES medium (DMEM-F12, 20% KSR, 1× Glutamax, 1× MEM-NEAA, 1× beta-mercaptoethanol, 0.2× penicillin/streptomycin, and 8 ng/ml FGF2). For transient reprogramming, cells were flow sorted at days 10, 13, 15, or 17 of reprogramming for the CD13+ SSEA4− and CD13− SSEA4+ populations. These cells were then replated on iMEFs (to replicate culture conditions before the flow sort and aid in cell reattachment) in fibroblast medium without doxycycline and then maintained like fibroblasts without iMEFs for subsequent passages. Cells were grown without doxycycline for 4 weeks in the first experiment and 5 weeks in the second experiment. Cells had returned to fibroblast morphology by 4 weeks in the second experiment, however, needed to be further expanded to generate enough material for downstream analyses. Negative control cells underwent the same procedure as the transient reprogramming cells to account for the effects of growing cells on iMEFs in hES media, flow sorting cells and keeping cells in culture for extensive periods of time. These confounders appeared to have no major effects on fibroblasts as these cells still clustered with the starting fibroblasts in our principal component analyses (*Figure 1—figure supplement 1D and E*). For complete reprogramming, colonies were picked on day 30 of reprogramming and transferred onto Vitronectin coated plates in E8 medium without doxycycline. Colonies were maintained as previously described (*Milagre et al., 2017*) and harvested at day 51 of reprogramming to ensure that the SP was completed and that traces of donor memory were erased. For Sendai virus iPSC reprogramming using CytoTune-iPS 2.0 Sendai Reprogramming Kit (Invitrogen), fibroblasts were reprogrammed as previously described (*Milagre et al., 2017*). For intermediate time points, cells were flow sorted into reprogramming (CD13− SSEA4+) and not reprogramming populations (CD13+ SSEA4−) before downstream profiling.

### Fluorescence-activated cell sorting of reprogramming intermediates

Cells were pre-treated with 10 µM Y-27632 (Stemcell Technologies) for 1 hr. Cells were harvested using StemPro Accutase cell dissociation reagent and incubated with antibodies against CD13 (PE, 301704, BioLegend), SSEA4 (AF647, 330408, BioLegend), and CD90.2 (APC-Cy7, 105328, BioLegend) for 30 min. Cells were washed two times with 2% FBS in phosphate-buffered saline (PBS) and passed through a 50-µm filter to achieve a single cell suspension. Cells were stained with 1 µg/ml DAPI just prior to sorting. Single color controls were used to perform compensation and gates were set based on the 'negative control intermediate' samples. Cells were sorted with a BD FACSAria Fusion flow cytometer (BD Biosciences) and collected for either further culture or DNA/RNA extraction.

## DNA methylation array

Genomic DNA was extracted from cell samples with the DNeasy Blood and Tissue Kit (QIAGEN) by following the manufacturer's instructions and including the optional RNase digestion step. For intermediate reprogramming stage samples, genomic DNA was extracted alongside the RNA with the AllPrep DNA/RNA Mini Kit (QIAGEN). Genomic DNA samples were processed further at the Barts and the London Genome Centre and run on Infinium MethylationEPIC arrays (Illumina).

## RNA-seq

RNA was extracted from cell samples with the RNeasy Mini Kit (QIAGEN) by following the manufacturer's instructions. For intermediate reprogramming stage samples and Sendai virus reprogrammed samples, RNA was extracted alongside the genomic DNA with the AllPrep DNA/RNA Mini Kit (QIAGEN). RNA samples were DNase treated (Thermo Fisher Scientific) to remove contaminating DNA. RNA-seq libraries were prepared at the Wellcome Sanger Institute and run on a HiSeq 2500 system (Illumina) for 50-bp single-end sequencing. For Sendai virus reprogrammed samples, libraries were prepared as previously described (*Milagre et al., 2017*), and run on a HiSeq 2500 (Illumina) for 75-bp paired-end sequencing.

## DNA methylation analysis

The array data was processed with the minfi R package and NOOB normalization to generate beta values. DNA methylation age was calculated using the multi-tissue clock (*Horvath, 2013*), the skin and blood clock (*Horvath et al., 2018*), the epiTOC clock (*Yang et al., 2016*), the GrimAge clock (*Lu et al., 2019a*), the Hannum clock (*Hannum et al., 2013*), the PhenoAge clock (*Levine et al., 2018*), and the Weidner clock (*Weidner et al., 2014*). We note that 19 CpG sites from the multi-tissue clock are missing in the Infinium MethylationEPIC array, however, the predictions are still robust when performed on NOOB normalised array data (*McEwen et al., 2018*). Telomere length was calculated using the telomere length clock (*Lu et al., 2019b*). Reference data sets for reprogramming fibroblasts and iPSCs were obtained from *Ohnuki et al., 2014* (GEO: GSE54848), *Banovich et al., 2018* (GEO: GSE110544), and *Horvath et al., 2018*. In addition, the reference data sets included novel data examining the intermediate stages of dermal fibroblasts being reprogrammed with the CytoTune-iPS 2.0 Sendai Reprogramming Kit (Invitrogen).

## RNA-seq analysis

Reads were trimmed with Trim Galore (version 0.6.2) and aligned to the human genome (GRCh38) with Hisat2 (version 2.1.0). Raw counts and log2 transformed counts were generated with Seqmonk (version 1.45.4). Reference data sets for fibroblasts and iPSCs were obtained from *Fleischer et al., 2018* (GEO: GSE113957) and *Banovich et al., 2018* (GEO: GSE107654). In addition, the reference data sets included novel data examining the intermediate stages of dermal fibroblasts being reprogrammed with the CytoTune-iPS 2.0 Sendai Reprogramming Kit (Invitrogen). Samples were carried forward for further analysis if they had a total read count of at least 500,000 with at least 70% of the reads mapping to genes and at least 65% of the reads mapping to exons.

## Immunofluorescence and imaging

Young control dermal fibroblasts were purchased from Lonza, Gibco, and the Coriell Institute (GM04505, GM04506, GM07525, GM07545, and AG09309) and were used at passage 4 after purchase. Antibody staining was performed as previously described (*Santos et al., 2003*) on cells grown on coverslips or cytospun onto coverslip after fixation with 2% paraformaldehyde for 30 min at room temperature. Briefly, cells were permeabilized with 0.5% TritonX-100 in PBS for 1 hr; blocked with 1% BSA in 0.05% Tween20 in PBS (BS) for 1 hr; incubated overnight at 4°C with the appropriate primary antibody diluted in BS; followed by wash in BS and secondary antibody. All secondary antibodies were Alexa Fluor conjugated (Molecular Probes) diluted 1:1000 in BS and incubated for 30 min. For the morphology analysis, cells were not permeabilized and were stained with direct labeled primary antibodies. Incubations were performed at room temperature, except where stated otherwise. DNA was counterstained with 5 µg/ml DAPI in PBS. Optical sections were captured with a Zeiss LSM780 microscope (63× oil-immersion objective). Fluorescence semi-quantification analysis was performed with Volocity 6.3 (Improvision). 3D rendering of z-stacks was used for semi-quantification of collagen I

and IV. Single middle optical sections were used for semi-quantification of H3K9me3. Antibodies and dilutions used are listed below:

> Anti-H3K9me3; 07-442, Merck/Millipore (1:500)
> Anti-Collagen I; ab254113, Abcam (1:400)
> Anti-Collagen IV; PA5-104508, Invitrogen (1:200)
> Anti-CD44-BB515; 564587, BD Biosciences (1:400)
> Anti-SSEA4-APC; FAB1435A-100; R&D Systems (1:40)
> Anti-CD13-PE; 301704; BioLegend (1:500)

## Wound healing assay

Cells were seeded into wound healing assay dish (80466, Ibidi) at a cell density of 20,000 cells per chamber. GM04505, GM04506, GM07545, and AG09309 fibroblasts were used at passage 5 as young controls. After 24 hr, the insert was removed generating 500 μm gaps between the cell-containing areas. The dishes were imaged every 20 min for 20 hr using a Nikon Ti-E equipped with a full enclosure incubation chamber (37°C; 5% $CO_2$) and the 20× objective. The images were pre-processed by cropping and rotating so that the wound area was on the right-hand side of the image. A Fiji macro was used to generate masks of the wound healing images. The coverage of the wound by immerging cells was analyzed by measuring the intensity of the mask along a line across the image. R was used to determine the location of the wound edge by collecting all of the x coordinates where the mask intensity was high enough to indicate that it was no longer part of the wound. The wound edge at each time point was expressed relative to the starting position to obtain the distance closed. Migration speed was calculated from the gradient between distance closed and time.

## Data analyses

Downstream analyses of RNA-seq and DNA methylation data were performed using R (version 4.0.2). Ggplot2 (version 3.3.2) was used to generate the bar charts, boxplots, line plots, pie charts, scatter plots, and violin plots. ComplexHeatmap (version 2.4.3) was used to generate the heatmaps. The combat function from the package sva (version 3.36.0) was used in **Figure 1E and F** to batch correct the novel Sendai reprogramming data set to the other data sets. The combat function was also used in **Figure 3** to batch correct the fibroblast aging reference data set (**Fleischer et al., 2018**) to our data set. Nonparametric tests were used when the data distribution was not normal and parametric tests were used when the data distribution was normal.

The random forest-based transcription clock was trained on the batch corrected aging reference data set using the caret R package (**Kuhn, 2008**) and random forest regression with tenfold cross validation, three repeats and a 'tuneLength' of 5. Chronological age was transformed before training with the following formulas adapted from the Horvath multi-tissue epigenetic clock (**Horvath, 2013**):

> $F(age) = \log_2(\text{chronological.age} + 1) - \log_2(\text{adult.age} + 1)$ if chronological.age ≤ adult.age
> $F(age) = (\text{chronological.age} - \text{adult.age}) / (\text{adult.age} + 1)$ if chronological.age > adult.age

As with the Horvath multi-tissue epigenetic clock, adult.age was set to 20 years old for these calculations (**Horvath, 2013**). The BiT age clock was also retrained on the batch corrected aging reference data set using scikit-learn as previously described (**Meyer and Schumacher, 2021**). This retrained model had a median absolute error of 5.55 years and consisted of 29 genes.

Rejuvenated CpG sites were found by comparing the methylation difference due to the age (calculated with the **Horvath et al., 2018** data set) to the methylation difference due to 13 days of transient reprogramming. CpG sites were classified as rejuvenated if they demonstrated a methylation difference of 10% over 40 years of aging that was reversed by transient reprogramming.

## Acknowledgements

The authors would like to thank all members of the Reik lab for helpful discussions. The authors would like to thank the bioinformatics facility at the Babraham Institute for processing the sequencing data, and the flow cytometry facility at the Babraham Institute for cell sorting. The authors would also like to thank the sequencing facilities at the Sanger Institute and the Bart's and the London Genome Centre for sequencing and methylation array services, respectively. This work was funded by the BBSRC. AP

is supported by a Sir Henry Wellcome Fellowship (215912/Z/19/Z). WR is a consultant and shareholder of Cambridge Epigenetix. TS is CEO and shareholder of Chronomics.

## Additional information

### Competing interests
Thomas M Stubbs: is CEO and shareholder of Chronomics. Wolf Reik: is a consultant and shareholder of Cambridge Epigenetix. Is employed by Altos Labs. The other authors declare that no competing interests exist.

### Funding

| Funder | Grant reference number | Author |
|---|---|---|
| Biotechnology and Biological Sciences Research Council | | Diljeet Gill<br>Fátima Santos<br>Hanneke Okkenhaug<br>Irene Hernando-Herraez<br>Thomas M Stubbs<br>Inês Milagre<br>Wolf Reik |
| Wellcome Trust | 215912/Z/19/Z | Aled Parry |
| Milky Way Research Foundation | | Diljeet Gill<br>Wolf Reik |
| Wellcome Investigator award | 210754/Z/18/Z | Wolf Reik<br>Christopher D Todd |

The funders had no role in study design, data collection and interpretation, or the decision to submit the work for publication.

### Author contributions
Diljeet Gill, Conceptualization, Data curation, Formal analysis, Investigation, Methodology, Visualization, Writing - original draft, Writing – review and editing; Aled Parry, Methodology, Writing – review and editing; Fátima Santos, Data curation, Formal analysis, Investigation, Writing – review and editing; Hanneke Okkenhaug, Formal analysis, Methodology; Christopher D Todd, Irene Hernando-Herraez, Investigation; Thomas M Stubbs, Conceptualization, Methodology, Data curation, Investigation, Supervision; Inês Milagre, Conceptualization, Data curation, Investigation, Methodology, Writing – review and editing; Wolf Reik, Conceptualization, Funding acquisition, Project administration, Supervision, Writing – review and editing

### Author ORCIDs

Diljeet Gill http://orcid.org/0000-0002-5725-2466
Aled Parry http://orcid.org/0000-0001-5192-3727
Fátima Santos http://orcid.org/0000-0002-3854-4084
Hanneke Okkenhaug http://orcid.org/0000-0003-0669-4069
Christopher D Todd http://orcid.org/0000-0003-2663-6173
Irene Hernando-Herraez http://orcid.org/0000-0003-1193-8419
Thomas M Stubbs http://orcid.org/0000-0002-2990-7381
Inês Milagre http://orcid.org/0000-0003-4753-5523
Wolf Reik http://orcid.org/0000-0003-0216-9881

### Decision letter and Author response
Decision letter https://doi.org/10.7554/eLife.71624.sa1
Author response https://doi.org/10.7554/eLife.71624.sa2

## Additional files

### Supplementary files
• Supplementary file 1. A comparison of previous transient reprogramming methods.

• Supplementary file 2. The complete results of the Tukey's range test that was used to compare the morphology ratio between the different stages and groups of MPTR.

• Supplementary file 3. The lists of fibroblast genes that were identified to either temporarily downregulate, temporarily upregulate or persist in their expression during MPTR.

• Transparent reporting form

### Data availability

DNA methylation array and RNA-seq data are available on Gene Expression Omnibus under the accession number: GSE165180.

The following dataset was generated:

| Author(s) | Year | Dataset title | Dataset URL | Database and Identifier |
|---|---|---|---|---|
| Gill D, Parry A, Santos F, Okkenhaug H, Todd CD, Hernando-Herraez I, Stubbs TM, Milagre I, Reik W | 2022 | Multi-omic rejuvenation of human cells by maturation phase transient reprogramming | https://www.ncbi.nlm.nih.gov/geo/query/acc.cgi?acc=GSE165180 | NCBI Gene Expression Omnibus, GSE165180 |

The following previously published datasets were used:

| Author(s) | Year | Dataset title | Dataset URL | Database and Identifier |
|---|---|---|---|---|
| Fleischer JG | 2018 | Predicting age from the transcriptome of human dermal fibroblasts | https://www.ncbi.nlm.nih.gov/geo/query/acc.cgi?acc=GSE113957 | NCBI Gene Expression Omnibus, GSE113957 |
| Banovich NE | 2018 | Impact of regulatory variation across human iPSCs and differentiated cells [RNA-seq] | https://www.ncbi.nlm.nih.gov/geo/query/acc.cgi?acc=GSE107654 | NCBI Gene Expression Omnibus, GSE107654 |
| Ohnuki M | 2014 | Critical role of transient activation of human endogenous retroviruses during reprogramming toward pluripotency | https://www.ncbi.nlm.nih.gov/geo/query/acc.cgi?acc=GSE54848 | NCBI Gene Expression Omnibus, GSE54848 |
| Banovich NE | 2018 | Impact of regulatory variation across human iPSCs and differentiated cells [methylation] | https://www.ncbi.nlm.nih.gov/geo/query/acc.cgi?acc=GSE110544 | NCBI Gene Expression Omnibus, GSE110544 |

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
