## [Editor Report]

This study describes a novel "maturation phase transient reprogramming" (MPTR) method to restore the epigenome of cells to a more youthful state. The authors demonstrate the effectiveness of the method to reverse several age-related changes including remodeling of the transcriptome. The method performs favorably compared to other transient reprogramming protocols, and the study will be of interest to developmental biologists as well as researchers who study ageing.

---

## [Decision Letter]

**Decision letter after peer review:**

Thank you for submitting your article "Multi-omic rejuvenation of human cells by maturation phase transient reprogramming" for consideration by *eLife*. Your article has been reviewed by 3 peer reviewers, and the evaluation has been overseen by a Reviewing Editor and Matt Kaeberlein as the Senior Editor. The reviewers have opted to remain anonymous.

Essential revisions:

1. It is discussed that fibroblast morphology is reversed. It would be good to quantify this morphological dynamics. For instance, whether cell size undergoes transition from mesenchymal to epithelial lineages and if any reversal is observed.

2. The data will strongly profit from one function test like wound healing or any other simple assay to analyze the function of the transiently reprogrammed fibroblast in comparison to the negative control for example.

3. Based on the standard protocols used for the culture of fibroblasts using culture medium containing fetal bovine serum (FBS), it is possible that the recovery of cellular identity following reprogramming is mainly due to differentiation signals coming from factors present in the medium. For these reasons, knockout serum (KSR) is used at later stages (day 8) of reprogramming to allow generation of iPSCs. The authors should rule out the possibility that recovery of fibroblast identity is due to the culture of reprogramed fibroblasts in FBS containing medium. For this purpose, the authors should test whether the fibroblast identity can be recovered following withdrawal by culturing the cell in KSR or 1% FBS containing medium instead of 10% FBS. This is a very important concept for the message that the manuscript tries to communicate regarding an epigenetic memory responsible for the recovery of fibroblast identity.

*Reviewer #2 (Recommendations for the authors):*

Concerns/Comments

I do not get the title (it might be the multi-omic…). Rejuvenation is achieved by Yamanaka, but not by multi-omics. So, the current title does not work.

There needs to be more information provided on the fibroblast. Passage numbers, expansions etc. That is only briefly mentioned in the MaM section. Passaging might influence aging….

The PC analysis of the samples is somewhat difficult to understand and actually not very informative nor fully convincing. What would happen simply without the reference data set in the PC analysis? This should be shown.

The intermediate cells fall indeed together with the reference cells doing almost exactly the same, 10-20 days of reprogramming. In that case, it might have been nice to have a fully reprogrammed set off these sample probes (expression of 40 days of the factors), and not simply a reference set.

Within the expression PC analysis, day 10-17 fall closely together, but not so in the CpG analysis. That might need to be commented on.

Is there are real difference in Figure 1d between the transient reprogramming intermediate and the failed to transiently reprogram intermediate? That might need to be the major focus of these analyses.

This reviewer does not appreciate picking out single, individual genes like in Figure 1e or g, as the overall global changes count, not single genes. Picking on single genes might be a bit misleading for the reader, especially as it is not clear whether these genes have a central function in the whole process.

It would further strengthen the manuscript if there was more information on the limitation of the transient procedure? At which length of reprogramming will we see additional negative effects on the overall procedure? Will they still return to be fibroblast after transient reprogramming for longer periods etc? That is not really addressed.

The overall question that Figure 2 is trying to address is indeed interesting.

The type of analyses provided in Figure 2 though remain very superficial, so that at the end the question is what additional novel and informative data does Figure 2 provide other than that the transient intermediate is not a fibroblast anymore, and that after stopping of Yamanaka, they return to become fibroblasts again, which though is already part of Figure 1. Again, out of a large number of genes, only 2 are picked that share a distinct pattern, but it is not listed how many other genes might share this pattern, and whether they might then also contribute to the phenotype, nor is there an attempt to validate the function of one or the other gene in the fibroblasts.

While the data in Figure 3 is really strong, there is concern about the conclusions of data from Figure 3c and Supplementary Figure 3c with respect to the optimum days of Yamanaka exposure for rejuvenation, as there is not analysis on differences among the transiently reprogrammed samples between day 10 and 17. That needs to be included to validate that statement in lines 350 and 360,361 of the manuscript (see also my comments above on limitations of the procedure).

The data will strongly profit from one function test like wound healing or any other simple assay to analyze the function of the transiently reprogrammed fibroblast in comparison to the negative control for example.

Data presented in Figure 4 is a bit redundant and for example epi-clock data is already part of Figure 1 and the new epic-clock data might be simply already included in Figure 1. For the H3K9me3 data, what is the statistics between failed to reprogram and reprogrammed? That is missing and interesting to know. The focus on the overlap of gene expression and epigenetics is highly interesting, and these analyses could be easily more expanded on, or some more information and context provided, as these genes might now be indeed more important.

Discussion lines 544 to 551. I am not sure whether that the data allows to compare directly extent of rejuvenation to other approaches, as distinct analyses have been done in these publications, and direct functional comparisons have not been done/performed. While obviously there is a great level of rejuvenation within the approach the authors introduced, whether that is substantially greater than xy might require more detail comparisons on multiple levels.

The translational aspects listed in line 571 to 574 is somewhat vague and need to be either described in more detail or simply omitted.

---

## [Author Response]

Essential revisions:1. It is discussed that fibroblast morphology is reversed. It would be good to quantify this morphological dynamics. For instance, whether cell size undergoes transition from mesenchymal to epithelial lineages and if any reversal is observed.

This is an interesting point. To address this, we have quantified the morphological changes using confocal microscopy and measuring a ratio of roundness (the maximum length divided by the perpendicular width) of individual cells before, during and after maturation phase transient reprogramming (MPTR). Cells became temporarily rounder during MPTR (lower ratio) and then returned to an elongated state (higher ratio) which matched that of the starting fibroblasts (Figure 1D).

2. The data will strongly profit from one function test like wound healing or any other simple assay to analyze the function of the transiently reprogrammed fibroblast in comparison to the negative control for example.

We agree that a functional measure would be very informative and so we have performed an in vitro wound healing assay to measure the migration speed of transiently reprogrammed fibroblasts and compared them to negative control fibroblasts as well as young control fibroblasts (Figure 3G). Negative control fibroblasts from middle-aged donors moved more slowly than young control fibroblasts into the scratch wound and transient reprogramming partially restored migration speed, suggesting some functional rejuvenation.

3. Based on the standard protocols used for the culture of fibroblasts using culture medium containing fetal bovine serum (FBS), it is possible that the recovery of cellular identity following reprogramming is mainly due to differentiation signals coming from factors present in the medium. For these reasons, knockout serum (KSR) is used at later stages (day 8) of reprogramming to allow generation of iPSCs. The authors should rule out the possibility that recovery of fibroblast identity is due to the culture of reprogramed fibroblasts in FBS containing medium. For this purpose, the authors should test whether the fibroblast identity can be recovered following withdrawal by culturing the cell in KSR or 1% FBS containing medium instead of 10% FBS. This is a very important concept for the message that the manuscript tries to communicate regarding an epigenetic memory responsible for the recovery of fibroblast identity.

The effect of FBS in promoting the return to fibroblast identity is an interesting possibility. We planned to investigate this by growing cells in fibroblast medium containing 10% KSR instead of 10% FBS after withdrawal of doxycycline following 13 days of reprogramming, as suggested. However, we have found that human fibroblasts were unable to be cultured long-term in KSR containing media. In addition, human fibroblasts grew substantially slower in 1% FBS containing medium (the other condition suggested), which would prevent us from collecting sufficient material with our current protocol. In addition, the substantially reduced growth speed would be a confounding factor that would limit the utility of any conclusions drawn. So unfortunately, whilst it was a very good suggestion, in practise it proved not possible to do these experiments.

**Author response image 1. sa2fig1:** 

Reviewer #2 (Recommendations for the authors):Concerns/CommentsI do not get the title (it might be the multi-omic…). Rejuvenation is achieved by Yamanaka, but not by multi-omics. So, the current title does not work.

For our title, we aimed to highlight that the rejuvenation is present in multiple-omic layers, and so we described this as multi-omic rejuvenation. This could be rephrased to “Maturation phase transient reprogramming promotes multi-omic rejuvenation in human cells”.

There needs to be more information provided on the fibroblast. Passage numbers, expansions etc. That is only briefly mentioned in the MaM section. Passaging might influence aging….

We tried to use the lowest passage number available to reduce the effect of in vitro culture on epigenetic age. Cells were used at passage four after purchasing and this has been added to the methods section (line 549). The exact passage number at purchase is unfortunately not available from Thermo Fisher.

The PC analysis of the samples is somewhat difficult to understand and actually not very informative nor fully convincing. What would happen simply without the reference data set in the PC analysis? This should be shown.

For the principal component analysis, the same trends are observed when the reference datasets are excluded with PC1 demonstrating the extent of reprogramming. As with the current figures, transiently reprogrammed fibroblasts cluster with the starting fibroblast samples and negative controls. We have added these additional PCA plots to Figure 1—figure supplement 1 (Figure 1—figure supplement 1D and 1E).

The intermediate cells fall indeed together with the reference cells doing almost exactly the same, 10-20 days of reprogramming. In that case, it might have been nice to have a fully reprogrammed set off these sample probes (expression of 40 days of the factors), and not simply a reference set.

We profiled fully reprogrammed cells which clustered with the reference iPSCs. We have added these samples along with the starting fibroblasts to Figures 1E and 1G.

Within the expression PC analysis, day 10-17 fall closely together, but not so in the CpG analysis. That might need to be commented on.

This is an interesting observation, which suggests that changes in the DNA methylome occur more gradually whereas changes in the transcriptome occur in more discrete stages. This has been commented on in the Results section (lines 176-180).

Is there are real difference in Figure 1d between the transient reprogramming intermediate and the failed to transiently reprogram intermediate? That might need to be the major focus of these analyses.

The transient reprogramming intermediate and failing to transiently reprogram intermediate samples cluster distinctly in figure 1E as well as in plots without the reference samples, supporting the idea that these populations are distinct. In addition, Nanog is only expressed in transient reprogramming intermediate cells and not the failing to transiently reprogram intermediate cells (Figure 1F).

This reviewer does not appreciate picking out single, individual genes like in Figure 1e or g, as the overall global changes count, not single genes. Picking on single genes might be a bit misleading for the reader, especially as it is not clear whether these genes have a central function in the whole process.

We have tried to represent RNA-seq data in such a way that global patterns are clear (PCA analyses, scatter plots etc.). We believe that representative examples of well-known genes such as Nanog, in addition to global analyses, improve clarity of the paper.

It would further strengthen the manuscript if there was more information on the limitation of the transient procedure? At which length of reprogramming will we see additional negative effects on the overall procedure? Will they still return to be fibroblast after transient reprogramming for longer periods etc? That is not really addressed.

We investigated several timepoints within the maturation phase in our study to determine the optimal amount of reprogramming for maximum rejuvenation. With 17 days of MPTR, we already observe diminished rejuvenation according to transcription and epigenetic clocks. We hypothesise that after the maturation phase, transient reprograming will be more difficult as the endogenous pluripotency factors are activated and so the reprogramming factors can no longer be reverted by withdrawing doxycycline. This point is now discussed in lines 118-120.

The overall question that Figure 2 is trying to address is indeed interesting.The type of analyses provided in Figure 2 though remain very superficial, so that at the end the question is what additional novel and informative data does Figure 2 provide other than that the transient intermediate is not a fibroblast anymore, and that after stopping of Yamanaka, they return to become fibroblasts again, which though is already part of Figure 1. Again, out of a large number of genes, only 2 are picked that share a distinct pattern, but it is not listed how many other genes might share this pattern, and whether they might then also contribute to the phenotype, nor is there an attempt to validate the function of one or the other gene in the fibroblasts.

Figure 2 highlights that some fibroblast genes remain expressed at high levels in transient reprogramming intermediate cells, whilst others are temporarily down-regulated but their enhancers remain lowly methylated. We have added the number of genes in each cluster to Figure 2—figure supplement 1B and provided the lists of gene names in supplementary file 3. The genes in figure 2E are examples from the clusters in Figure 2—figure supplement 1B.

While the data in Figure 3 is really strong, there is concern about the conclusions of data from Figure 3c and Supplementary Figure 3c with respect to the optimum days of Yamanaka exposure for rejuvenation, as there is not analysis on differences among the transiently reprogrammed samples between day 10 and 17. That needs to be included to validate that statement in lines 350 and 360,361 of the manuscript (see also my comments above on limitations of the procedure).

Transcriptional clocks suggest that transiently reprogrammed cells with 17 days of reprogramming are older than transiently reprogrammed cells with shorter lengths of reprogramming. This is also in line with the observations from epigenetic clocks (Figure 4B and Figure 4—figure supplement 1B).

The data will strongly profit from one function test like wound healing or any other simple assay to analyze the function of the transiently reprogrammed fibroblast in comparison to the negative control for example.

We agree that a functional measure would be very informative and so we have performed an in vitro wound healing assay to measure the migration speed of transiently reprogrammed fibroblasts and compared them to negative control fibroblasts as well as young control fibroblasts (Figure 3G). Negative control fibroblasts from middle-aged donors moved more slowly than young control fibroblasts into the scratch wound and transient reprogramming partially restored migration speed, suggesting some functional rejuvenation.

Data presented in Figure 4 is a bit redundant and for example epi-clock data is already part of Figure 1 and the new epic-clock data might be simply already included in Figure 1. For the H3K9me3 data, what is the statistics between failed to reprogram and reprogrammed? That is missing and interesting to know. The focus on the overlap of gene expression and epigenetics is highly interesting, and these analyses could be easily more expanded on, or some more information and context provided, as these genes might now be indeed more important.

The aim of figure 4 is to demonstrate the rejuvenation at the epigenome level following the process of maturation phase transient reprogramming (MPTR), including by epigenetic clock analyses. The epigenetic clock analysis in figure 1 is different in that it does not include cells that have undergone MPTR, rather epigenetic clocks are used across a reprogramming time course to define the timepoint when rejuvenation occurs.

For figure 4A, we have added statistics comparing failed to transiently reprogram and transiently reprogrammed cells.

We agree that the overlap between epigenetic and transcriptional rejuvenation is interesting, and we have expanded upon the roles of these genes in the Results section (lines 422-430).

Discussion lines 544 to 551. I am not sure whether that the data allows to compare directly extent of rejuvenation to other approaches, as distinct analyses have been done in these publications, and direct functional comparisons have not been done/performed. While obviously there is a great level of rejuvenation within the approach the authors introduced, whether that is substantially greater than xy might require more detail comparisons on multiple levels.

We appreciate that this is an important caveat. Indeed, we have re-analysed some public data generated using a different transient reprogramming method using our transcriptome clock (Figure 3—figure supplement 1b) and found that our method results in more substantial rejuvenation in comparison, but we agree that a more direct and thorough comparison using identical readouts will be necessary to confirm this. We have discussed these points in the Discussion section (lines 458-461).

The translational aspects listed in line 571 to 574 is somewhat vague and need to be either described in more detail or simply omitted.

We have increased the amount of detail provided for potential translational aspects by discussing the potential of MPTR to rejuvenate cells for autologous cell transplants and provided a potential example (for treating skin wounds). We have also elaborated on the potential for our method to form the basis of a screen (lines 490-496).